# Local and landscape-level diversity effects on forest functioning

**Jacqueline Oehri**[1]*, **Marvin Bürgin**[1], **Bernhard Schmid**[1,2], **Pascal A. Niklaus**[1]*

1 Department of Evolutionary Biology and Environmental Studies, University of Zurich, Zurich, Switzerland,
2 Department of Geography, University of Zurich, Zurich, Switzerland

* jacqueline.oehri@ieu.uzh.ch (JO); pascal.niklaus@ieu.uzh.ch (PAN)

**Data Availability Statement:** All relevant data are within the paper and its Supporting Information files.

**Funding:** This study was funded by the University of Zurich Research Priority Program Global Change and Biodiversity (URPP GCB) grant to Pascal A.

## Abstract

Research of the past decades has shown that biodiversity is a fundamental driver of ecosystem functioning. However, most of this biodiversity–ecosystem functioning (BEF) research focused on experimental communities on small areas where environmental context was held constant. Whether the established BEF relationships also apply to natural or managed ecosystems that are embedded in variable landscape contexts remains unclear. In this study, we therefore investigated biodiversity effects on ecosystem functions in 36 forest stands that were located across a vast range of environmental conditions in managed landscapes of Central Europe (Switzerland). Specifically, we approximated forest productivity by leaf area index and forest phenology by growing-season length and tested effects of tree species richness and land-cover richness on these variables. We then examined the correlation and the confounding of these local and landscape-level diversity effects with environmental context variables related to forest stand structure (number of trees), landscape structure (land-cover edge density), climate (annual precipitation) and topography (mean altitude). We found that of all tested variables tree species richness was among the most important determinants of forest leaf area index and growing-season length. The positive effects of tree species richness on these two ecosystem variables were remarkably consistent across the different environmental conditions we investigated and we found little evidence of a context-dependent change in these biodiversity effects. Land-cover richness was not directly related to local forest functions but could nevertheless play a role via a positive effect on tree species richness.

## Introduction

Biodiversity-ecosystem functioning (BEF) studies of the past decades have revealed that biodiversity is an important driver of ecosystem functioning [1]. Most studies were concerned with effects on primary productivity, which embodies the efficiency of a community in capturing essential resources and converting these into biomass [2]. To date, BEF studies have predominantly focused on experimental plant communities that were established on relatively small spatial extents and in relatively homogenous environments [3, 4]. This allowed researchers to

Niklaus and Gabriela Schaepman-Strub. The funders had no role in study design, data collection and analysis, decision to publish, or preparation of the manuscript.

**Competing interests:** The authors have declared that no competing interests exist.

systematically identify causal effects of biodiversity, with productivity generally linearly increasing with the logarithm of species richness [1].

The BEF relationships identified in experiments largely encompass effects of local biodiversity that are driven by local mechanisms. To which extent the BEF relationships found in experiments also apply to so-called "real-world" [5] ecosystems in natural or managed landscapes remains unclear [3, 4, 6].

Natural and managed landscapes provide ecosystem functions and services of paramount importance to humans [7]. In these landscapes, ecosystems are connected by the exchange of energy, matter and organisms and thus form meta-ecosystems [8–10]. In such meta-ecosystems, conditions at the local scale are strongly intertwined with the patterns and processes at larger scales of space, time and biological organization [11–13]. In other words, real-world ecosystems are embedded in a larger-scale environmental context that typically is variable, unlike the situation usually realized in small-scale BEF experiments. This larger-scale context may give rise to larger-scale mechanisms that potentially affect local biodiversity and ecosystem functions [3, 14]. For example, species may disperse in the environment and colonize habitats in which they otherwise would not occur, thereby increasing local diversity and ecosystem functioning [13, 15, 16]. The larger-scale environmental context affects not only spatial and demographic processes of plants but also other taxa including consumers and pathogens [17–19]. Environmental context variables that are especially important in this respect are the connectivity of habitats, land use, climate, topography and the spatial heterogeneity of these variables [13, 14, 20–22]. All this complexity is purposely excluded from the plot-level experiments on which our current understanding of BEF relationships largely rests.

In summary, we argue that it has become increasingly important to investigate BEF relationships in real-world conditions such as human-dominated landscapes, which represent around 76% of the terrestrial ice-free biosphere [23]. Such studies of real-world BEF relationships need to consider the larger-scale environmental context of ecosystems, in addition to local species diversity. Here, we did so by studying leaf area index as an indicator of productivity and growing-season length as a measure of phenology in 36 forest stands across a broad range of environmental conditions in Switzerland over a period of two years. Both leaf area index and growing-season length capture important, complementary aspects of forest functioning: leaf area index approximates the amount of leaves in a forest, which are important for the present magnitude of forest productivity [24–26] and forest growing-season length approximates the yearly duration of forest productivity [27–29]. The investigated forest stands were systematically selected to cover three major forest types in Central Europe: coniferous, broadleaf and mixed. The study design was replicated in the six distinct biogeographic regions of Switzerland; these differ with respect to the regional species pool, climatic, topographic and edaphic conditions [30]. We quantified the extent to which our forest functioning metrics could be explained by a range of local and landscape-scale variables. At the local scale, we used species richness as the primary predictor of ecosystem functioning because it is the key design variable in BEF experiments. In analogy, at the landscape scale, we focused on land-cover richness as a predictor. We then extended our statistical models with metrics characterizing the environmental context in terms of local forest stand structure, landscape structure, climate and topography. For each of these environmental categories, we assessed several candidate variables and finally selected one representative variable: Specifically, we approximated forest stand structure by the number of tree individuals in the stand, landscape structure by the edge density metric ([31]; the relative amount of interfaces between land-cover types), climate by annual precipitation and topography by mean altitude. Using all these variables we tested the following hypotheses: 1) local- and landscape-level diversity (i.e. tree species richness and land-cover richness) explain forest functions, 2) the amount of variation in forest functions

explained by these diversity metrics is important relative to the amount of variation explained by environmental variables and 3) diversity effects on forest functions do not or only weakly interact with environmental variables.

## Materials and methods

### Study design

Using 36 forest stands, we analyzed the relationship of local ecosystem functioning measures to local and landscape-level variables characterizing diversity and environmental conditions. These forest stands were a subset of 1416 sites that form an observation network of the Swiss biodiversity monitoring program (BDM, Z9 plot network, biodiversitymonitoring.ch; [32]; Fig 1). The 36 selected sites formed a complete block design with the six biogeographic regions (BGR; [30]) as blocks and the three forest types (FT) coniferous, broadleaf and mixed within each block. In order to classify the BDM sites into coniferous, broadleaf, mixed and non-forest, we calculated Jaccard similarities [33] between the sets of plant species characteristic for the 30 typical forest communities of Switzerland (S1 Table; [34]) and the sets of plant species monitored at each of the 1416 BDM sites. We then randomly picked two replicates per forest type and block, with the constraint that the sites were accessible. Based on the tree inventory we later conducted at each site, six of the 36 forests were reclassified and biogeographic region and forest type therefore no longer fully orthogonal. At each site, we established a circular inventory plot with an area of 950 m$^2$.

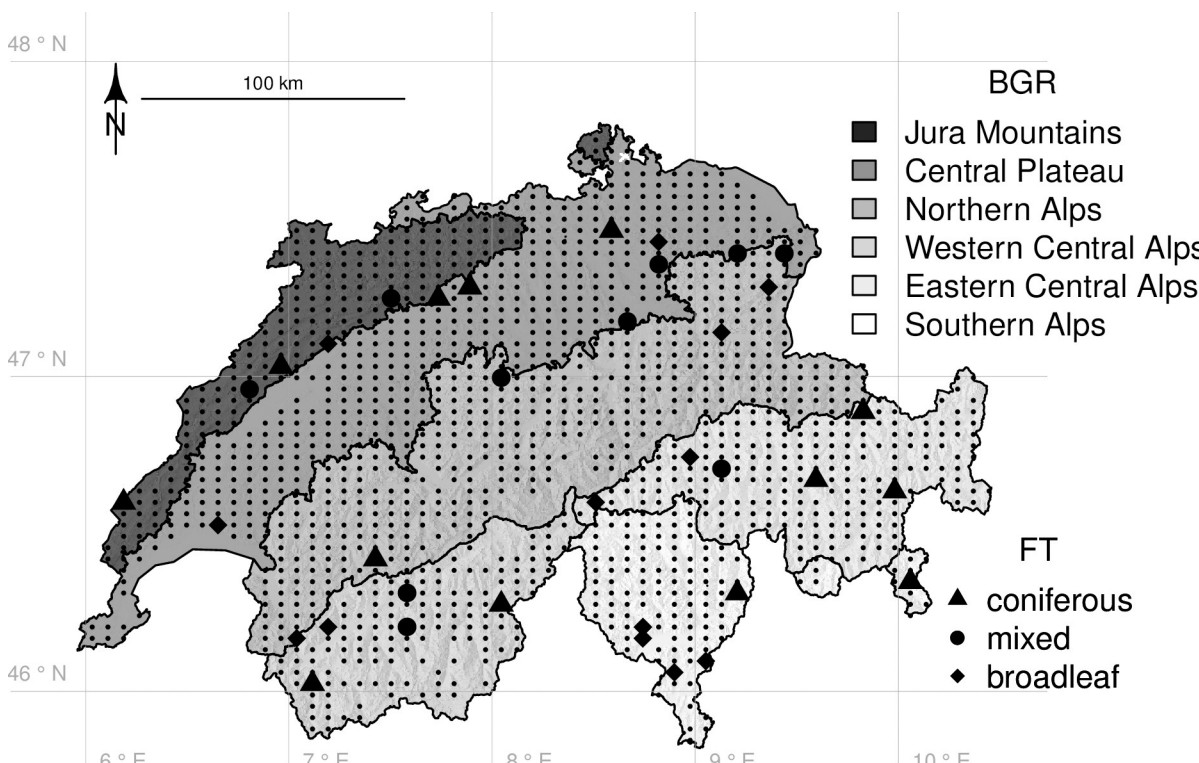

**Fig 1. Study design.** From a total of 1416 sites that are part of the Swiss biodiversity monitoring program (BDM Z9 plot network; biodiversitymonitoring.ch) and regularly spread across Switzerland (small black dots) we selected 36 sites that we classified as "coniferous", "mixed" or "broadleaf" forests based on the Jaccard similarity of BDM-monitored plant communities (S1 Table; [34]). We selected two replicates of each forest type (FT) in each of the six biogeographic regions (BGR; [30]). We re-classified six forest types after executing the tree inventories, so that the final distribution of forest types is not fully orthogonal with biogeographic region (triangles = coniferous forests, n = 13; circles = mixed forests, n = 10; diamonds = broadleaf forests, n = 13).

## Local tree diversity and stand structure

We marked all trees in each inventory plot with a diameter at breast height (DBH) of at least 5 cm and determined their species identity. The inventory revealed the presence of 2341 tree individuals belonging to 33 species (S2 Table). Using these data, we determined tree density ($n_{trees}$: number of trees per plot), tree species richness (SR), rarefied tree species richness (scaled to the plot with the smallest number of trees: n = 21), plot-level total stem basal area (BA) and an indicator combining stand age and demographic structure, the average DBH of the three largest trees ($DBH_{max}$).

## Landscape-level diversity, structure, climate and topography

We determined land-cover composition around our plots using land-cover classifications of points arranged on a 100m rectangular grid (product NOAS04, Swiss Federal Statistical Office, GEOSTAT). We aggregated the original 17 land-cover types into the classes forest, meadow, urban, arable, water, urban green, bare land (e.g. scree slopes) and unproductive (e.g. ruderal areas). Using this grid data, Fragstats [31] and the ClassStat/PatchStat functions in the R library SDMTools [35], we determined land-cover richness (LR; the number of different land-cover types in a landscape), patch density (PD; the number of distinct patches of land-cover in a landscape) and edge density (ED; the total length of interface between different land-cover types divided by the total area of a landscape) in circular areas with radii varying from 100–6400 m. The variable LR had the broadest range among landscapes of 600 m radius and we therefore used this radius for the calculation of all our landscape-level metrics. We further calculated a measure of forest connectivity ($F_{conn}$; named "cohesion" in Fragstats) and the fractional cover of forest in the landscape ($F_{frac}$).

We characterized landscape-level topography by mean altitude (alt; range: 331–1940 m above sea level), slope (range: 4.6–39.9 degrees) and the northerly aspect of the slope (N-aspect; range: -0.85 to 0.84; negative values for south, positive values for north) based on a digital elevation model (product DHM25, Swiss Federal office of Topography: swisstopo). As indicators of climate, we used average mean annual temperature (temp; range: 1.1–12.5˚C) and annual precipitation (precip; range: 802–1934 mm; years 2000–2016, products TabsM and RhiresM, Swiss Federal office of Meteorology and Climatology: MeteoSwiss).

## Local ecosystem functioning

**Productivity.** We determined leaf area index (LAI), defined as the plant canopy leaf area per unit ground area [24–26]. LAI is a widely used proxy of vegetation productivity [24–26].

To determine LAI we recorded digital hemispheric photographs (DHP; [26, 36]) at three random locations in each inventory plot once in summer 2015 and 2016, respectively, (average date of visit: 28. August). The camera (Nikon D90, Sigma 4.5mm f/2.8 circular fish eye lens, ISO 200) was mounted on a tripod 1 m above ground pointing vertically upwards. To achieve even illumination of the sky, we took the photos under overcast conditions or at dusk or dawn. DHPs were pre-processed to correct for remaining gradients in sky brightness and LAI calculated with Hemisfer 2.1 [37] with algorithms correcting for slope [38] and clumping [39]. We averaged the three LAI measurements per plot in each year and then across years.

**Phenology.** An important aspect of forest phenology [40] is the yearly growing-season length (GSL), which refers to the yearly period of elevated forest activity (note that plant growing season could differ from climatological growing season [27]).

We assessed forest growing-season length as the yearly period of elevated canopy leaf area that we derived from the seasonal change in light attenuation by the forest canopy. Specifically, in each inventory plot, we installed an automatic light logger (HOBO UA-002-64; Onset Computer

Corporation, Bourne, MA) 40 cm above ground with the sensor facing vertically upwards. A second logger was placed close to the plot but outside the forest canopy where no objects obstructed the light measurement. We fitted the loggers with a short-pass filter (cut off at 716 nm, KG1 716FHC6565, Knight Optical Ltd., Kent, UK) to record only photosynthetically active radiation (PAR; 400–700 nm). The sensors were calibrated in the laboratory using an integrating sphere (RTS-3ZC; ASD Inc., Longmont, CO) and a spectroradiometer (FieldSpec 4 Standard-Res; ASD Inc., Longmont, CO). Illuminance (lux) was recorded half hourly from spring 2015 until autumn 2017 (S1 Fig). We determined the attenuation of PAR as one minus the ratio of daily average inside to outside recordings (11–16h). These time series were smoothed by fitting a cubic spline (15 degrees of freedom) before determining start (SOS) and end (EOS) of the growing season in every year. Following established guidelines from land surface phenology research [41, 42], we defined SOS as date at which illuminance attenuation first exceeded the mean of annual minimum and maximum. Conversely, EOS was determined as final date at which attenuation remained above this threshold. Growing-season length (GSL) was calculated as EOS minus SOS (S1 Fig).

## Data analysis

We first assessed the overall correlation coefficients (r) among variables characterizing local and landscape-level diversity, landscape structure, topography, climate, stand structure and forest ecosystem functioning. We evaluated the strength of these correlations according to [43], where effects are categorized as small, medium and large for r = 0.1, r = 0.3 and r = 0.5, respectively. We used these overall correlations, together with a principal component analysis (S2 Fig) to select a set of relatively independent variables that characterized important environmental gradients. As an indicator of landscape structure, we used edge density (ED), which was highly correlated with patch density (PD), forest connectivity ($F_{conn}$) and forest fraction ($F_{frac}$; S2 Fig). As an indicator of topography, we selected altitude (alt), which was indicative also of the climate variable mean annual temperature (temp). We kept precipitation (precip) as a climate variable because it was relatively independent of altitude and temperature (S2 Fig). Finally, we used the number of trees ($n_{trees}$) in the plot to characterize local forest stand structure. Because we could adjust for $n_{trees}$ in our models, we focused on species richness instead of rarefied species richness in our analyses. Untransformed and log-transformed tree species richness [log(SR)] resulted in virtually identical fits; here, we present results for log(SR), which we preferred because many meta-analyses have shown that productivity and other ecosystem-functions typically increase linearly with log(SR) [1]. As an indicator of landscape-level diversity, we used land-cover richness (LR). We always kept the factors biogeographic region (BGR) and forest type (FT) in our analyses models because they were the basis of our plot selection.

Using these eight explanatory variables (i.e. log(SR), LR, ED, alt, precip, $n_{trees}$, BGR and FT), we analyzed the variation in the dependent variables LAI and GSL, our metrics for ecosystem functioning. We assessed the predictive power of the explanatory variables by variance partitioning (method by [44] as implemented in the lmg function of the R library relaimpo [45]). To avoid overfitting, we assessed only models with a maximum of three predictors. Specifically, we created models with all possible subsets of 1, 2 and 3 out of the eight explanatory variables. For each such model, we quantified the variance explained by the predictors when averaged over all possible orderings in the model ("lmg" method), when fitted first ("first" method) and when data were first adjusted for all other predictors ("last" method) using the variance partitioning method [45]. We then averaged the results that were first averaged for models with 1, 2 and 3 predictors separately.

Finally, we assessed the confounding of effects of log(SR) and LR with environmental context variables. Therefore, using general linear models summarized by analysis of variance (ANOVA;

R library ASReml-R [46]; VSN International, Hemel Hemsted), we tested effects of log(SR) and LR after fitting (i.e. accounting for the effects of) each of the other selected predictors separately. We then repeated the analyses with these models that additionally fitted the study design variable biogeographic region (BGR) first. Using these models, we determined the effect sizes for log(SR) and LR as partial correlations derived from F-ratios: [47, 48]. If not indicated differently, data processing and analyses were done using the software R 3.5 (http://r-project.org).

## Results

### Correlations among local and landscape-level predictors and ecosystem functions

We found positive correlations of local tree species richness (log-transformed) with local forest functioning measured by leaf area index (r = 0. 38, P<0.05) and, at a lower significance, with growing-season length (r = 0.38, P<0.1; Fig 2). Log-transformed tree species richness was also positively related to land-cover richness in the surrounding landscape (r = 0.39, P<0.05) but land-cover richness did not relate significantly to leaf area index or growing-season length. Besides tree species richness, the only other variable in our set that correlated significantly with leaf area index was precipitation (r = 0.37, P<0.05). In the case of growing-season length, we found strong correlations only with temperature and altitude (r = 0.80 and -0.83, respectively, P<0.01 for both).

The landscape structure variables edge density, patch density, the fraction of forested area and the connectivity of forest patches formed a highly correlated complex (|r|>0.47, P<0.01 for all pairs; Fig 2). Interestingly, these variables correlated with land-cover richness (range |r| = 0.45–0.69, all P<0.01, except connectivity of forest patches: r = -0.22, n.s.) but were independent of tree species richness. In other words, land-cover richness captured some essence of both tree species richness and landscape structure.

Of the topographic and climatic variables (Fig 2), altitude and temperature were highly correlated. Expectedly, temperature decreased with altitude (r = -0.86, P<0.001) and was lower in landscapes with more north-exposed and steep slopes (r = -0.35, and r = -0.34, both P<0.05). Precipitation was not significantly related to these variables.

Investigating the relationships among local stand structure variables, we found that the size of the largest trees ($DBH_{max}$), which we consider a proxy of stand age, correlated positively with plot basal area (r = 0.72, P<0.001) and negatively with the number of trees (r = -0.59, P<0.001).

There was little correlation among variables of different groups (delimited with bold black lines in Fig 2). Exceptions were log-transformed tree species richness, which correlated positively with temperature and negatively with altitude (r = 0.48, and r = -0.50, respectively, both P<0.01) and the number of trees, which increased with land-cover richness, edge density and patch density (range r = 0.33–0.4, all P<0.05).

### Variance explained by local and landscape-level predictors

Models containing 1–3 of the eight explanatory variables relating to diversity, environmental context and study design explained on average 16.2% of variation in leaf area index and 35.1% of variation in growing-season length (measured by the coefficient of determination $R^2$).

In the case of leaf area index, tree species richness (log-transformed) was among the most important predictors and explained, on average, 12.9% of the overall variance (Fig 3; $R^2$ averaged across models with 1–3 predictors). Depending on sequential order, i.e. for which variables the effect was adjusted first, log-transformed tree species richness explained 11.9–14.0%

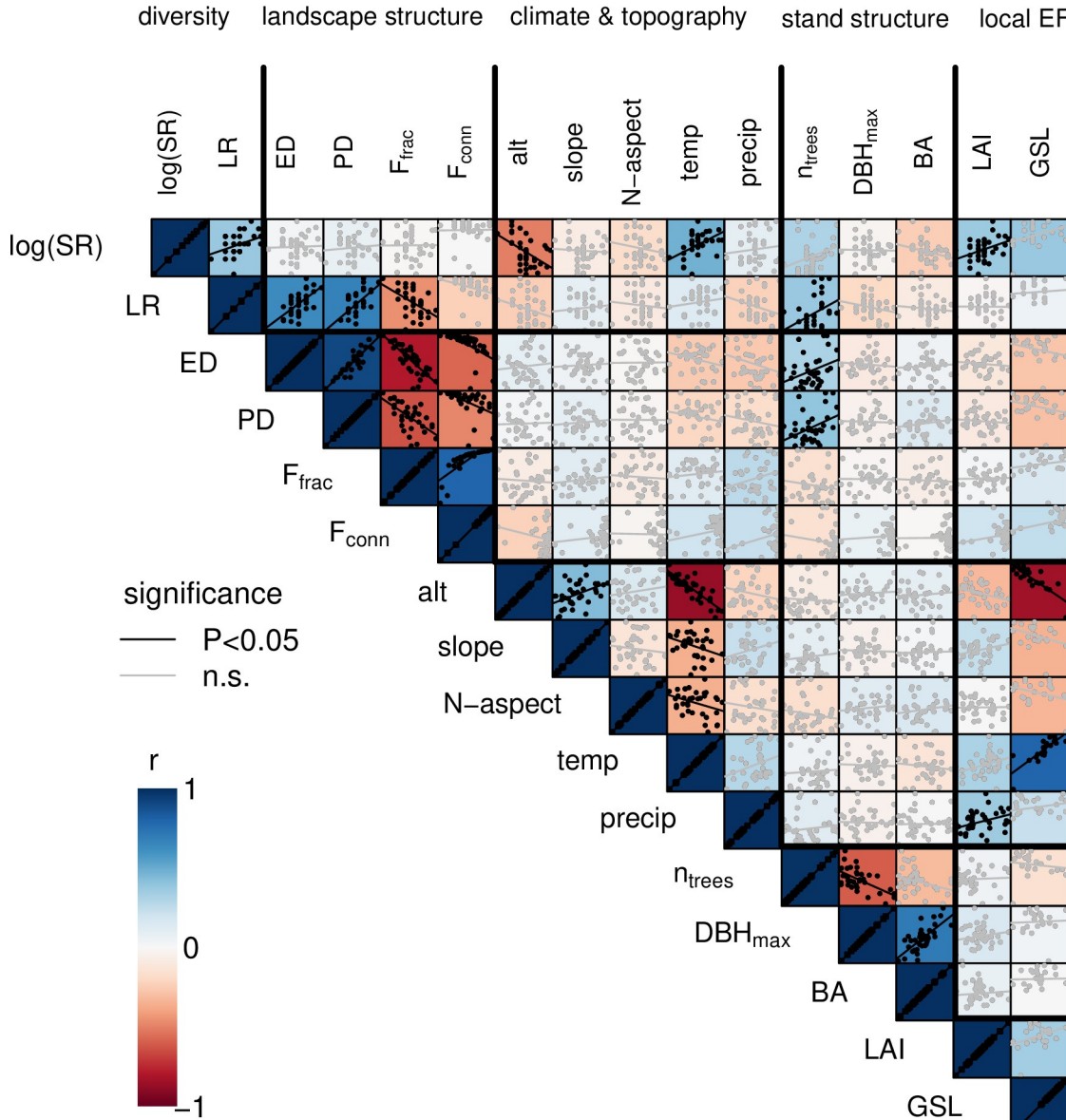

**Fig 2. Correlations among local and landscape-level predictors and local ecosystem functions.** Significance (P<0.05) is indicated in black, sign of correlation is indicated with red to blue coloring (blue = positive correlation, red = negative correlation). Bold black lines delimit groups of variables characterizing local and landscape-level diversity, landscape structure, climate and topography, forest stand structure and ecosystem functioning (local EF). log(SR): log-transformed tree species richness; LR: land-cover richness; ED: edge density; PD: patch density; $F_{frac}$: fraction of forested area; $F_{conn}$: connectivity of forest patches; alt: altitude; N-aspect: northerly aspect; temp: temperature; precip: precipitation; $n_{trees}$: number of trees, $DBH_{max}$: average DBH of three largest trees; BA: cumulative stem basal area in the forest stand. Number of study units n = 36 for leaf area index (LAI) and n = 22 for growing-season length (GSL).

of the overall variance ($R^2$ averaged across models with 1–3 predictors). The only variable exceeding the predictive power of tree species richness was the study design variable biogeographic region (average $R^2$: 25.2%; $R^2$ averaged across models with 1–3 predictors).

A near-identical picture was present for growing-season length, with a similar amount of variance explained by the logarithm of tree species richness (average $R^2$: 13.6%; range of $R^2$: 12.6–14.5%; $R^2$ averaged across models with 1–3 predictors). The main difference in the

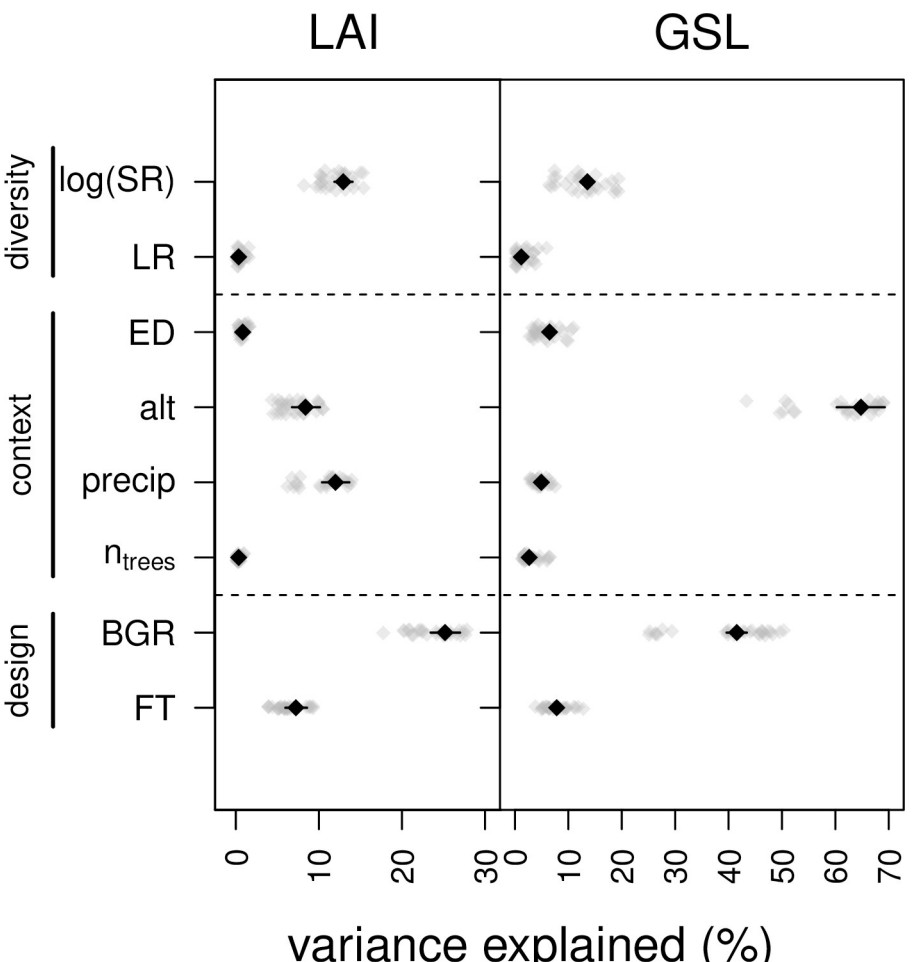

**Fig 3. Variance in local forest functions explained by diversity, environmental context and study design variables.** The coefficients of determination ($R^2$) for models that include all combinations of 1–3 of all 8 predictors (listed on y-axis) are on average 16.2% for leaf area index (LAI) and 35.1% for growing-season length (GSL), respectively. The $R^2$ contributions of every predictor when averaged over all possible model-orderings are indicated for each model with light-grey dots. Results that were averaged across all models with 1–3 predictors are shown in black. Black dots represent average $R^2$ contributions over all possible model-orderings and black lines indicate the range of $R^2$ contributions when included first or last in the model, respectively. log(SR): log-transformed tree species richness; LR: land-cover richness; ED: edge density; alt: altitude; precip: precipitation; $n_{trees}$: number of trees, BGR: biogeographic region; FT: forest type. Number of study units n = 36 for LAI and n = 22 for GSL.

analysis of growing-season length compared with the analysis of leaf area index was that altitude and biogeographic region explained a very large fraction of the overall variance (average $R^2$: 64.8% and 41.5%, respectively; ranges of $R^2$: 60.3–69.2% and 39.6–43.4%, respectively; $R^2$ averaged across models with 1–3 predictors). This is to be expected because growing-season length strongly depends on temperature.

The logarithm of tree species richness explained more variance in leaf area index and growing-season length than the number of trees (average $R^2$ LAI: 0.3%; GSL: 2.7%), forest type (average $R^2$ LAI: 7.2%; GSL: 7.8%) and precipitation (average $R^2$ LAI: 12.0%; GSL: 4.9%; Fig 3; $R^2$ averaged across models with 1–3 predictors).

Land-cover richness, landscape structure represented by edge density and stand structure represented by the number of trees only explained a marginal amount of variance and were not important in any model (Fig 3).

## Species and landscape diversity effects, adjusted for environmental context

We calculated effect sizes ($r_{partial}$) of tree species richness (log-transformed) in separate models that each first adjusted for (i.e. fitted) the effect of one of the environmental context variables. Using models explaining leaf area index, we found effect sizes of tree species richness (log-transformed) that were relatively large ($r_{partial} > 0.3$) and similar in magnitude independent of the specific models (range $r_{partial}$: 0.36–0.41), except when these first adjusted for altitude ($r_{partial}$: 0.26) and forest type ($r_{partial}$: 0.28; Table 1).

In the case of growing-season length, log-transformed tree species richness effect sizes were relatively similar in magnitude in most models (range $r_{partial}$: 0.31–0.41) but strongly decreased after adjusting for altitude ($r_{partial}$: 0.13) and strongly increased after adjusting for biogeographic region ($r_{partial}$: 0.66; Table 1).

Importantly, high $r_{partial}$ values (>0.3) for log-transformed tree species richness were restored when the models additionally adjusted for the study design variable biogeographic region (i.e. model classes 10–16 with "BGR+..." in the model in Table 1). Effect sizes of log-transformed tree species richness in these models always exceeded 0.3 in the case of leaf area index (range of $r_{partial}$: 0.33–0.41; Table 1). Tree species richness effects on growing-season length in these models were even larger than the effects on leaf area index (range $r_{partial}$: 0.47–0.66; Table 1).

Effect sizes for land-cover richness generally were low and insignificant for leaf area index and growing-season length (Table 1), although they increased in models of growing-season

**Table 1. Effects of local and landscape-level diversity on forest functions when fitted after environmental context variables.**

| class | model terms | species richness [log(SR)] effect | | | | landscape richness (LR) effect | | | |
|---|---|---|---|---|---|---|---|---|---|
| | | LAI | | GSL | | LAI | | GSL | |
| | | $r_{partial}$ | P-value | $r_{partial}$ | P-value | $r_{partial}$ | P-value | $r_{partial}$ | P-value |
| 1 | - | **0.38** | **0.024** | **0.38** | 0.080 | -0.01 | 0.970 | 0.07 | 0.759 |
| 2 | log(SR) | - | - | - | - | -0.18 | 0.312 | 0.00 | 0.985 |
| 3 | LR | **0.41** | **0.015** | **0.38** | 0.093 | - | - | - | - |
| 4 | BGR | **0.40** | **0.025** | **0.66** | **0.004** | 0.11 | 0.568 | **0.34** | **0.182** |
| 5 | FT | 0.28 | 0.114 | **0.31** | **0.179** | -0.09 | 0.627 | -0.04 | 0.882 |
| 6 | ED | **0.38** | **0.025** | **0.34** | 0.136 | 0.07 | 0.692 | 0.28 | 0.222 |
| 7 | alt | 0.26 | 0.126 | 0.13 | 0.585 | -0.09 | 0.612 | 0.06 | 0.796 |
| 8 | precip | **0.36** | **0.031** | **0.38** | 0.093 | 0.08 | 0.642 | 0.04 | 0.854 |
| 9 | $n_{trees}$ | **0.38** | **0.025** | **0.41** | 0.064 | -0.03 | 0.869 | 0.12 | 0.601 |
| 10 | BGR + log(SR) | - | - | - | - | -0.06 | 0.760 | **0.31** | **0.236** |
| 11 | BGR + LR | **0.39** | **0.032** | **0.65** | **0.007** | - | - | - | - |
| 12 | BGR + FT | **0.33** | 0.084 | **0.65** | **0.009** | 0.04 | 0.852 | 0.24 | 0.385 |
| 13 | BGR + ED | **0.40** | **0.028** | **0.66** | **0.005** | 0.12 | 0.518 | **0.43** | **0.097** |
| 14 | BGR + alt | **0.41** | **0.023** | **0.47** | 0.068 | 0.05 | 0.792 | 0.20 | 0.460 |
| 15 | BGR + precip | **0.38** | **0.037** | **0.65** | **0.006** | 0.11 | 0.553 | **0.35** | **0.179** |
| 16 | BGR + $n_{trees}$ | **0.38** | **0.039** | **0.65** | **0.007** | 0.06 | 0.736 | **0.33** | **0.217** |

We calculated effect sizes ($r_{partial}$) for local tree species richness [log-transformed; log(SR)] and land-cover richness (LR) in linear models in which we first fitted the environmental context variables as indicated in the column "model terms" (e.g. models of class 1 contain only log(SR) or LR as predictors, respectively, models of class 2 contain log(SR) and LR as predictors and models of class 10 contain the three predictors BGR, log(SR) and LR). We separately conducted these analyses for the dependent ecosystem functioning variables leaf area index (LAI) and growing-season length (GSL). High $r_{partial}$ values (>0.3) are indicated with bold typeface, significance is indicated in the column "P-value" and the direction of the relationship is indicated with positive or negative sign before the $r_{partial}$ value. Cells with "-": no model fitted; BGR: biogeographic region; FT: forest type; ED: edge density; alt: altitude; precip: precipitation; $n_{trees}$: number of trees. Number of study units n = 36 for leaf area index and n = 22 for growing-season length.

length when additionally adjusting for biogeographic region. In these models the $r_{partial}$ of land-cover richness ranged from 0.31–0.43, except in the cases where we first adjusted for altitude ($r_{partial}$: 0.20) or forest type ($r_{partial}$: 0.24; Table 1).

## Discussion

We investigated the effects of local and landscape-level diversity and environmental context variables on forest functioning in managed landscapes. We found that local tree diversity, measured as the logarithm of species richness, was positively related to forest leaf area index and growing-season length. Tree diversity was among the most important predictors for these functions compared with other variables related to landscape structure (land-cover edge density), climate (annual precipitation), topography (mean altitude) and stand structure (number of trees). Local tree diversity effects were relatively robust, with little confounding with the environmental context variables. Overall, our findings thus support all three hypotheses stated in the Introduction in the case of local tree diversity, and suggest that it is an important driver of forest productivity and phenology, with effects that remain relatively constant across the range of environmental conditions encountered in managed landscapes. In contrast, landscape-level land-cover richness did not show a strong direct relationship with forest productivity and phenology and hence, we found no support for the hypotheses stated in the Introduction in the case of landscape diversity. However, land-cover richness strongly positively correlated with local tree species richness. It may thus be that landscape diversity contributes to local ecosystem functioning indirectly through effects mediated by local tree species diversity.

The positive effects of tree species diversity on forest leaf area index that we found expand previous findings from experimental [49–52] and non-experimental studies [53] and support theoretical expectations [54] that mixed-species forests might have increased ecosystem functioning due to efficient "canopy packing" or "crown complementarity". These effects of canopy packing can be predicted if co-existing species differ in crown architecture either genetically or via neighborhood driven plastic responses in crown growth and vertical leaf distribution [49, 53, 55].

We found no significant relationship between tree species richness and stand basal area—if anything, this relationship tended to being negative (correlation coefficient r = -0.24, n.s., Fig 2), contrasting previous evidence from local-scale comparative studies [56–58]. Whether local diversity effects on basal area were truly absent or whether unaccounted drivers masked these, remains unclear. Swiss forests are often managed by removing individual trees from stands, avoiding clear-cutting large areas. It may be that differences in management history among plots had a long lasting effect on stand basal area, whereas leaf area recovered faster from such interventions. Independent of these considerations, our results suggest that productivity measured by leaf area index does not necessarily reflect productivity measured by woody biomass and that these two attributes are likely governed by different mechanisms. Hence, species richness effects on woody biomass production [59–61] might differ from species richness effects on the leaf area in forest stands.

In our study, tree species richness was positively correlated with both leaf area index and growing-season length. Correlation coefficients were similar for both variables but the relationship was only marginally significant for growing-season length. The reason for the lower statistical power in the case of growing-season length is that only 22 of the 36 forest sites showed a seasonal pattern in light attenuation (S1 Fig). Altitude and biogeographic region were dominant predictors of growing-season length, which strongly decreased with mean altitude ($F_{1,20} = 45.0$, P<0.001) and tended to be lower in the Western and Eastern Central Alps than in the other biogeographic regions ($F_{5,16} = 2.5$, P = 0.078). Although species richness was

not related to biogeographic region ($F_{5,30} = 0.5$, $P = 0.8$), it did decrease with altitude ($F_{1,34} = 11.1$, $P = 0.002$), indicating that species richness and altitude were confounded. This raised the possibility that effects of tree species richness on growing-season length might actually have been altitude effects in disguise. However, we found that the variation in growing-season length across biogeographic regions masked tree diversity effects. These increased substantially in magnitude and significance after we accounted for biogeographic region. Our findings of positive plot-level species richness effects on growing-season length parallel our earlier findings for mixed ecosystems (including large amounts of non-forest ecosystems) at the landscape scale [62]. Phenology plays an important role for species interactions [63] and the capability of communities to adapt to environmental change [28, 64]. Hence, biodiversity might be important for the resilience of communities faced with global change under real-world conditions in complex landscapes.

There is conflicting evidence whether landscape context is important for local ecosystem functioning. In agricultural grasslands and crop fields, landscape heterogeneity can increase local biological control [13, 16]. In tropical dry forest, landscape structure was less important for local aboveground biomass [65]. In the present study, land-cover richness did not directly affect our two measured local ecosystem functions. However, the close correlation between land-cover richness and local tree species richness indicated that land-cover richness may have had a indirect effect on local ecosystem functioning via an effect on tree species richness. This would be in accordance with the idea that environmental heterogeneity at larger scales drives local species richness [66]. Indirect effects of landscape diversity on local ecosystem functioning via positive effects on local biodiversity have indeed been proposed in theoretical studies [15, 67] and are empirically supported in grassland and agricultural areas [13, 16–19] as well as in forests [65, 68]. Possible mechanisms include that landscape diversity increases the regional taxonomic or functional diversity via an increased range of spatially dissimilar environmental conditions [66]. Increased regional diversity could promote local biodiversity and ecosystem functions via the spatial insurance effect [67], by which local ecosystems may be colonized by species with well-suited traits, or by species that would otherwise go locally extinct [15]. For example, it has been found that adverse effects of local land-use intensification on biodiversity and ecosystem functions were masked by the spillover of species from the surrounding landscape [13, 16].

In our study, the effects of log-transformed tree species richness stayed relatively constant when adjusting for different environmental context variables at the local and the landscape scale. This suggests that environmental context does not alter the local relationship between biodiversity and ecosystem functioning. This reasoning is supported by the fact that we did not find significant interactions between log-transformed tree species richness and environmental context variables on leaf area index or growing-season length when we included such interaction terms in further exploratory analysis, using the general linear models described in the Materials and methods section. The only exceptions were two significant interactions between log-transformed species richness and forest type or the number of trees on growing-season length. However, these interactions were essentially due to coniferous forest plots that had low numbers of trees and for which season length was ill-defined. Generally, the potential dependence of biodiversity effects on environmental context remains poorly tested to date [3, 6]. Experiments suggest that there is no such dependency ([69, 70]; but see [71]), whereas observational studies often suggest the contrary [59–61, 72–75]. However, cause and effect of diversity were often not separated in those latter studies.

To conclude, we show that local tree species diversity is a powerful predictor of local forest functions in managed real-world landscapes, similarly to the effects of species richness that have been found in BEF experiments. Landscape diversity had only a low explanatory power, but was positively correlated with local tree species diversity and could thus indirectly affect

local forest functions. A general challenge in observational studies is to disentangle causes and effects of biodiversity [6]. Whereas knowing the causes of biodiversity enables the identification of conditions that facilitate biodiversity, it is only knowledge about the effects of biodiversity that enables an assessment of the consequences of biodiversity loss for ecosystem functioning and ultimately, human well-being.

## Supporting information

**S1 Fig. Processing of light data. A**: Temporal development of average daily illuminance (kilolux; i.e. klx) measured inside (dark grey) and outside (light grey) forest stands using corresponding sensors (HOBO UA-002-64; Onset Computer Corporation, Bourne, MA) from spring 2015 (March, April, May; MAM) until autumn 2017 (September, October, November; SON). **B:** Light attenuation by the forest canopy derived from the smoothed ratio of average daily (i.e. between 11.00 am and 4.00 pm) inside and outside forest illuminances (light grey) subtracted from 1 (dark grey). We defined yearly start of the season (SOS) as the day of the year when light attenuation first exceeded the mean of its annual minimum and maximum value (horizontal dashed line). Similarly, we derived the yearly end of the season (EOS) as the last day of the year before light attenuation fell below this threshold. We then calculated yearly growing-season length (GSL) as the number of days between SOS and EOS and finally averaged GSL across the years 2015 and 2016. We derived these metrics only for forests with a clear seasonal pattern in light attenuation, which we determined by applying the following restrictions: A minimum GSL of 60 days, a minimum growing season amplitude of 0.045 and a maximum number of 2 periods of consecutive days above the mean of annual minimum and maximum light attenuation, with a maximum number of 8 days in between these periods. In total, we obtained GSL values for a subset of 22 out of the 36 forest sites.
(TIF)

**S2 Fig.** Principal component analyses (PCA) of variables related to landscape structure (A), climate and topography (B) and local stand structure (C). A: Edge density (ED; total length of borders between different land-cover types divided by the total landscape area), patch density (PD; the number of different land-cover patches divided by the total area of the landscape) and land cover richness (LR; the number of different land-cover types) closely clustered together and strongly differed in their loadings compared to the fraction of forest cover (Ffrac) and the connectivity of forest patches (Fconn) for the first principal component (PC1). B: Temperature (temp) and altitude (alt) showed the strongest negative covariance in loadings for PC1, indicating a strong temperature-altitude gradient in our dataset. We identified a second important gradient defined by precipitation (precip) that positively co-varied with slope and negatively co-varied with the northerly aspect (N-aspect) in their loadings for the second principal component (PC2). C: Our proxy for stand age and demographic structure DBHmax (average diameter at breast height of the three largest trees) clustered closely with cumulative basal area (BA) and both of these measures co-varied negatively with the number of trees (ntrees) in their loadings for PC1
(TIF)

**S1 Table. Forest communities found in Switzerland.** We used information on the typical species composition of the 30 main Swiss forest types [34] to associate 1416 sites of the Swiss biodiversity monitoring program (BDM; biodiversitymonitoring.ch; Fig 1) with a likelihood of being a i) coniferous, ii) broadleaf or iii) mixed forest by using presence data of vascular plant species (Z9 Indicator of the BDM) and Jaccard's index of similarity.
(DOCX)

**S2 Table. Tree inventory.** List of all the tree species we found in the tree inventory, the number of individuals across all study sites and number of study sites with a respective species present.
(DOCX)

**S1 Dataset.**
(ZIP)

# Acknowledgments

Andy Hueni helped with the calibration of the light sensors. We acknowledge Daniel Trujillo and Richard Baxter for their help with collecting field data. We thank the Federal Office for the Environment (FOEN) for the provision of the Swiss biodiversity monitoring program (BDM) data and acknowledge the dedicated botanists who conducted fieldwork for the BDM.

# Author Contributions

**Conceptualization:** Jacqueline Oehri, Bernhard Schmid, Pascal A. Niklaus.

**Data curation:** Jacqueline Oehri, Marvin Bürgin, Pascal A. Niklaus.

**Formal analysis:** Jacqueline Oehri, Marvin Bürgin, Pascal A. Niklaus.

**Funding acquisition:** Pascal A. Niklaus.

**Investigation:** Jacqueline Oehri, Marvin Bürgin, Pascal A. Niklaus.

**Supervision:** Pascal A. Niklaus.

**Visualization:** Jacqueline Oehri.

**Writing – original draft:** Jacqueline Oehri, Pascal A. Niklaus.

**Writing – review & editing:** Jacqueline Oehri, Marvin Bürgin, Bernhard Schmid, Pascal A. Niklaus.

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
