## [Decision Letter · Decision Letter 0]

4 Mar 2020

PONE-D-20-01220

Local and landscape-level diversity effects on forest functioning

PLOS ONE

Dear Dr. Oehri,

Thank you for submitting your manuscript to PLOS ONE. After careful consideration, we feel that it has merit but does not fully meet PLOS ONE’s publication criteria as it currently stands. Therefore, we invite you to submit a revised version of the manuscript that addresses the points raised during the review process.

ACADEMIC EDITOR:Please consider the comments on your manuscript,especially you should pay attention to the models and criteria used,make more explanations on the methods adopted.

We would appreciate receiving your revised manuscript by Apr 18 2020 11:59PM. To enhance the reproducibility of your results, we recommend that if applicable you deposit your laboratory protocols in protocols.io, where a protocol can be assigned its own identifier (DOI) such that it can be cited independently in the future. For instructions see: http://journals.plos.org/plosone/s/submission-guidelines#loc-laboratory-protocols

We look forward to receiving your revised manuscript.

Kind regards,

RunGuo Zang

Academic Editor

PLOS ONE

Additional Editor Comments (if provided):

Both reviewers are positive to your valuable work,however they raise some concerns on the clarifications of the methods and the functioning criteria.Please carefully consider theirsuggestions and revise the manuscript accordingly.

Journal Requirements:

Reviewers' comments:

Reviewer's Responses to Questions

**Comments to the Author**

1. Is the manuscript technically sound, and do the data support the conclusions?

Reviewer #1: Partly

Reviewer #2: Yes

2. Has the statistical analysis been performed appropriately and rigorously? 

Reviewer #1: No

Reviewer #2: Yes

3. Have the authors made all data underlying the findings in their manuscript fully available?

Reviewer #1: Yes

Reviewer #2: No

4. Is the manuscript presented in an intelligible fashion and written in standard English?

Reviewer #1: Yes

Reviewer #2: Yes

5. Review Comments to the Author

Reviewer #1: This is a well-conceived study; I like the idea of testing ideas about richness effects on ecosystem functioning in real world landscapes. In general, the study was designed well, but I am concerned that some of the models may be overfit, given the small sample size and number of predictor variables.

The manuscript was very well written and easy to understand in most places. The English expression is good but could be improved in a few spots (I haven’t made any comments about this).

Below are some specific comments

Line 104. I don’t understand the reference to Jaccard similarities – what do they have to do with the vegetation communities outlined in Table S1?

Line 140. If you’ve used average altitude doesn’t the variable reflect average altitude rather than topography (ie, I’m suggesting perhaps renaming it mean altitude)

Line 180-181. Is the PCA necessary? Does it add anything?

Line 183-185. I’d suggest using a model with poisson errors, as species richness is a count. Then your inferences are about species richness rather than log(SR). If you use a poisson model check for overdispersion.

Lines 195-210. It seems that at least some of your models contain all eight predictor variables. With a sample size of 36 (and I think 22 for GSL – line 233 – although I missed this in the methods) your models will be overfitted. A common rule of thumb is one estimated parameter for each 10 data points. While I think we can push this a bit, with 36 data points you could perhaps build a model with 4 or 5 continuous predictors. And, because BGR and FT are categorical, they make this problem worse. For example, BGR has 6 categories, so 5 parameters are estimated by the regression model. Ideally we’d like 40-50 samples to build a model containing only BGR. Unless I’ve misunderstood how you’ve analysed your data I fear your sample size is inadequate for the complexity of your models. Perhaps you could achieve your objectives using simpler models.

Line 212-221. If this section is about correlations among predictors (as the sub-heading suggests, remove LAI and GSL, as these are your response variables). As a general comment, although it’s interesting to know which of your original suite of predictors correlated with each other, I think it is your least important result, and I suggest putting it last in the results section. Lead with your most interesting result. If you proceed with reordering your sections also do so in the methods and discussion.

Line 258-259. I’m not sure what you mean by ‘ranged among the most important predictors’.

Line 335-338. You could answer this question using some kind of network analysis (eg a structured equation model or a Bayesian network).

Line 380-389. I don’t understand your point about a scale mismatch. Isn’t the idea behind this kind of analysis that landscape patterns influence processes occurring at a point? Perhaps I’m misunderstanding your meaning.

Reviewer #2: The manuscript "Local and landscape-level diversity effects on forest functioning" by Oehri et al. presents an assessment of the relationship between tree species richness and landcover-type richness on two measures of forest ecosystem function, leaf area index and growing season length, within 36 forest stands in Switzerland. The design of the study and the analyses of the data appear to be appropriate. The manuscript is well-written and describes results that provide further evidence of the importance of biodiversity to the functioning of forested ecosystems.

The manuscript has a few relatively minor shortcomings that should be addressed by the authors.

First, the Introduction could be improved by including a more specific statement of the hypotheses and/or objectives of the study. As written, the manuscript comes across as being somewhat descriptive, but I don't think that was the intent of the authors. Listing specific hypotheses in the Introduction and then addressing in the Discussion whether the results support these hypotheses would make for a stronger manuscript.

Second, I could not find an explanation for why the authors choose these particular ecosystem functions and indicators for this study. Why are these functions important, and why use these specific indicators to assess these functions?

Third, some important details are missing that would help readers to better understand the methods. For example, how were the locations of the sites determined? The caption for figure 1 (lines 109-110) says they were "regularly spread" across the country, but how? How was peak-season LAI determined (line 150)? The total peak over the two years? Or the mean of the two years? Or something else? I get the sense that LAI measurements were taken once a year at each plot. How did the authors determine when they would collect these measurements? How was principal component analysis (lines 180-181) used to verify the strengths of the correlations among the variables? How were the GLMs conducted to account for the effect of biogeographic region (lines 205-207)?

Fourth, some elaboration in the Discussion would be helpful about the relationships between tree species richness, growing season length and biogeographic regions (lines 367-372). How was biogeographic region controlled for? Are biogeographic region and growing season length related in some way?

Additional comments

Abstract:

-- The abstract makes no mention about where this study was conducted. This should be mentioned (possibly near lines 33-35).

-- Line 40: I don't think the word "ranges" works here. The word "is" works better.

-- Lines 39-41: Is it worth mentioning that growing season length and and tree species richness may be confounded with altitude?

Introduction:

-- Lines 52-54: This sentence is missing something.

-- Line 53: What is a "positive decelerating response"? Is there a clearer way to state this?

-- Lines 56-58: Note there have been a number of BEF studies in the United States and Canada using many thousands of plots in natural forest settings. See, for example:

Paquette, A. and Messier, C. 2011. The effect of biodiversity on tree productivity: from temperate to boreal forests. Global Ecology and Biogeography, 20: 170-180. doi:10.1111/j.1466-8238.2010.00592.x

Potter, K.M., and C.W. Woodall. 2014. Does biodiversity make a difference? Relationships between species richness, evolutionary diversity, and aboveground live tree biomass across U.S. forests. Forest Ecology and Management. 321:117-129. https://doi.org/10.1016/j.foreco.2013.06.026.

Iannone, B.V., K.M. Potter, K-A. Dixon Hamil, W. Huang, H. Zhang, Q. Guo, C.M. Oswalt, C.W. Woodall, and S. Fei. 2016. Evidence of biotic resistance to invasions in forests of the Eastern USA. Landscape Ecology. 31:85-99. https://doi.org/10.1007/s10980-015-0280-7.

-- Lines 80ff: Articles are needed in several cases in this paragraph, such as "an indicator", "a measure", "the primary indicator", etc.

Methods

-- Line 123: It's is probably worth noting here that the rarefied tree species richness was scaled to the plot with the smallest number of trees.

-- Line 163: "to only record" would be easier to read as "to record only"

-- Line 173: Should "about the threshold" be "above the threshold"?

-- Line 179: I recommend adding "categorized as" before "small, medium and large"

-- Line 184ff: "an" should be added before "indicator"

-- Line 198: Would it be possible to add a citation for the variance partitioning method?

Results

-- Line 215: I don't think the word "marginally" works here, because the correlation is actually fairly large while the level of significance is not. Something like "at a lower significance" or "at a higher P value"

Discussion

-- Line 349: I would change "was" to "may have been"

-- Line 362: "was" needed after "relationship"

-- Lines 392-395: This is an interesting idea. What would the mechanism be?

-- Lines 404-406: Should this issue with coniferous forest plots be addressed in the methods or results?

-- Lines 413-415: I think this idea could be elaborated on further, either here or elsewhere in the Discussion.

-- Line 418: "the" is needed before "effects"

6. PLOS authors have the option to publish the peer review history of their article (what does this mean?). If published, this will include your full peer review and any attached files.

Reviewer #1: No

Reviewer #2: Yes: Kevin M. Potter

---

## [Author Response · Author response to Decision Letter 0]

18 Apr 2020

All responses are contained within the "Response to Reviewers" document.

We thank both reviewers for their constructive reviews, which enabled us to improve our manuscript.

---

## [Editor Report · Decision Letter 1]

29 Apr 2020

Local and landscape-level diversity effects on forest functioning

PONE-D-20-01220R1

Dear Dr. Oehri,

We are pleased to inform you that your manuscript has been judged scientifically suitable for publication and will be formally accepted for publication once it complies with all outstanding technical requirements.

With kind regards,

RunGuo Zang

Academic Editor

PLOS ONE
---

## [Editor Report · Acceptance letter]

1 May 2020

PONE-D-20-01220R1 

Local and landscape-level diversity effects on forest functioning 

Dear Dr. Oehri:

I am pleased to inform you that your manuscript has been deemed suitable for publication in PLOS ONE. Congratulations! Your manuscript is now with our production department. 

With kind regards,

on behalf of

Professor RunGuo Zang 

Academic Editor

PLOS ONE